# DOLAMA 200: Effectiveness and Safety of a Dual Therapy with Dolutegravir Plus Lamivudine in Treatment-Experienced HIV-1 Infected Real World Participants in Spain

**DOI:** 10.3390/v16020259

**Published:** 2024-02-06

**Authors:** Sergio Sequera-Arquelladas, Carmen Hidalgo-Tenorio, Luis López-Cortés, Alicia Gutiérrez, Jesús Santos, Francisco Téllez, Mohamed Omar, Sergio Ferra-Murcia, Elisa Fernández, Rosario Javier, Coral García-Vallecillos, Juan Pasquau

**Affiliations:** 1Unit of Infectious Diseases, Instituto de Investigación Biosanitario de Granada (IBS-Granada), Virgen de las Nieves University Hospital, 18014 Granada, Spain; maria.hidalgo.tenorio.sspa@juntadeandalucia.es (C.H.-T.); charojavier@hotmail.com (R.J.); gvcoral@gmail.com (C.G.-V.); jpasquau@gmail.com (J.P.); 2Department of Infectious Diseases, Virgen del Rocio University Hospitals, 41013 Seville, Spain; luisfernando@lopezcortes.net (L.L.-C.); alicia.gutierrez.valencia@gmail.com (A.G.); 3IBIMA Plataforma BIONAND, Unit of Infectious Diseases, Virgen de la Victoria University Hospital, 29010 Málaga, Spain; med000854@gmail.com; 4Unit of Infectious Diseases, Puerto Real Universitary Hospital, 11510 Cádiz, Spain; francisco.tellez.perez@gmail.com; 5Unit of Infectious Diseases, Hospital Complex of Jaen, 23007 Jaén, Spain; omarampa@gmail.com; 6Unit of Infectious Diseases, Hospital Torrecárdenas Hospital, 04009 Almería, Spain; serfemu@gmail.com; 7Internal medicine Service, Hospital Poniente, 04700 Almería, Spain; elisaffuertes@gmail.com

**Keywords:** HIV, simplification, ART, dolutegravir, lamivudine, cost-effective

## Abstract

The continuous pharmacological advances in antiretroviral treatment (ART) and the increasing understanding of HIV drug resistance has led to a change in the paradigm of ART optimization in the setting of the viral suppression of treatment-experienced patients with the emerging evidence of the effectiveness and safety of dual therapies. The aim of this study is to determine the antiviral efficacy and safety of switching to Dolutegravir + Lamivudine in people living with HIV, and to analyze the rate of patients with virologic failure (VF). A total of 200 patients were included with a median age of 51 years, 189 cells/µL of nadir CD4+, 13 years on ART and four previous ART regimens. Among the 168 patients who completed a follow-up at 48 weeks, a total of five VFs occurred, resulting in a 2.98% (5/168) VF rate. The results of the intention-to-treat analysis were a VF rate of 2.54% (5/197), and the rate of patients/year with viral suppression was 98.3% (298/303) in the observed data analysis. We observed a significant improvement in mean CD4 lymphocytes, the CD4/CD8 ratio and lipid profiles. The optimization of ART to DTG plus 3TC is a cost-effective switch option for treatment-experienced HIV patients, and also improves their lipid profiles.

## 1. Introduction

The high efficacy achieved with the continuous pharmacological advances in antiretroviral therapy (ART) has led to a radical change in the prognosis of HIV infection. This brings the life expectancy of people living with HIV (PLWHIV) much closer to that of the general population [1].

These advances in antiviral efficacy and safety (short-term measurable toxicity, tolerance and convenience) have been demonstrated in a multitude of studies maintaining triple therapy as the gold standard of ART, and as the most widely used therapy worldwide [2]. Nowadays double therapy (2DR) is recognized as an elective option even for naïve patients [2,3,4]. Double therapy could also provide several benefits in treatment optimization from simplifying the regimen, reducing the pill burden, to enhance tolerability or decreasing long-term toxicity.

However, there are two pathogenic factors in this chronic and still incurable condition, which requires lifelong treatment at the moment, that will make it difficult to match life expectancy with that of the general population: (1) the long-term cumulative toxicity of antiretrovirals (ARVs) [5] and (2) residual viremia (detectable HIV-1 RNA below the threshold of ultrasensitive commercial viral load assays). The latter sustains a certain degree of immune activation and inflammation despite the maximum control of viral replication achieved by ART. [6] Both factors have the capacity to interfere with, and enhance, the aging process and the emergence of comorbidities, i.e., ‘Non-AIDS events’. These non-AIDS events have become the main cause of morbidity and mortality among PLWHIV [7,8,9].

The consolidated use of three ARVs in a chronic condition also generates important economic burdens and cost-effectiveness problems [9].

Hence, for many years, with the availability of ARVs with high antiretroviral activity and a high genetic barrier to the emergence of resistance, the search for therapeutic simplification strategies has been promoted. These simplification strategies maintain the same antiviral efficacy, have fewer ARVs, less toxic potential and a greater (or equal) capacity to control inflammation [10].

In this regard, dual therapy with dolutegravir (DTG) and lamivudine (3TC) has been able to demonstrate, through multiple clinical trials, non-inferiority to triple therapy in a wide range of possible antiretroviral treatment scenarios (from first-line use in naive patients to simplification in pre-treated patients and from double-blind randomized clinical trials to real-world observational studies) [11,12,13,14,15,16]. The advantages and benefits of this dual therapy in terms of cumulative toxicity and cost-effectiveness in the long-term treatment of PLWHIV [13] makes the communication of experiences and analyses of the efficacy, safety and cost-effectiveness of this dual therapy in real-life studies (for which the population will differ from those of clinical trials, and represent a wider and a more realistic scenes) contribute to reinforcing and consolidating the data from previous clinical trials and current guidelines.

## 2. Individuals and Methods

### 2.1. Study Design and Setting

This is a retrospective, observational, single-treatment-arm, multicenter study, with a follow-up of 48 weeks, to determine the antiviral efficacy and safety of switching to DTG (50 mg/QD) plus 3TC (300 mg/QD)in PLWHIV. Informed consent was obtained from all subjects involved in the study at the HIV infection follow-up clinics of the participating hospitals from October 2014 to December 2018. The study was completed through the collaboration of 8 hospitals, all from Andalucía (Spain).

The main objective was to analyze the rate of patients with virologic failure (2 consecutive viral loads ≥ 50 copies/mL) in the 48 weeks of follow-up and to determine the effectiveness of this dual therapy (3TC + DTG) by calculating the proportion of patients with virologic success, defined as 48-weeks follow-up without virologic failure (RNA HIV-1 ≥ 50 copies/mL at two consecutive determinations).

Viral suppression is defined as the last recorded viral load < 50 copies/mL (with the study combination therapy and without virologic failures during the follow-up)

The following secondary objectives were to analyze: (1) the rate of patients who reached week 24 and 48 with successful virological control; (2) the incidence and frequency of new adverse effects (AEs) after ART change; (3) the evolution of lymphocyte subpopulations and the CD4/CD8 ratio; (4) changes in lipid profiles (TC, LDL, HDL, TG, TC/HDL), renal function (creatinine), liver function (GOT, GPT, GGT, FA), metabolic parameters (calcium, phosphorus, vitamin D) and inflammatory biomarkers (C-Reactive Protein). These were recorded when available. Differences in the incidence of therapeutic failures between the baseline-24-week period and the 24–48-week period were also analyzed.

### 2.2. Inclusion and Non-Inclusion Criteria

Inclusion criteria: PLWHIV > 18 years old, with at least 6 months of ART; virological suppression maintained in the 6 months prior to the switch (1 blip allowed, defined as a viral load ≥50 copies/mL followed by a retest result of viral load <50 copies/mL); switched to DTG + 3TC for any reason. Patients with viral load ≥50 copies/mL detected in the baseline laboratory test results obtained at the moment of the switch will still be considered eligible if treatment with DTG + 3TC was maintained

Non-inclusion criteria: PLWHIV with active AIDS during the study period or hepatitis B virus (HBV) coinfection requiring treatment with tenofovir (HBV infection in effective treatment with lamivudine and entecavir was not a non-inclusion criterion); pregnancy; previous presence of HIV genotypes with resistance mutations to lamivudine or dolutegravir; patients who had experienced adverse effects related to DTG or 3TC.

### 2.3. Statistics and Outcomes

For the main objective analysis, patients were subset into 4 different subgroups: (1) intention-to-treat (ITT) analysis of all enrolled patients; (2) modified intention-to-treat (mITT) analysis, which excluded patients who withdrew from the study due to problems that altered the assessment of the predefined objectives, such as physician decision, patient decision (both always unrelated to adverse effects) and non-related deaths. AEs, VF and loss to follow-up maintained; (3) per protocol (PP) analysis, or patients who remained on the study combination until they reached 48 weeks or discontinued the study due to virologic failure or adverse effects (this subset will not include, in addition to mITT criteria, the loss to follow-up criterion); and (4) observed data (OD) analysis, of available virologic data for all patients during exposure to DTG + 3TC expressed in patients/year.

All data were collected in accordance with the current Spanish data protection law (Ley Orgánica 3/2018 de 5 de diciembre de protección de datos de carácter personal) and the study was approved by the Granada drug research ethics committee.

All the following data were acquired from the electronic clinical records of the selected patients:

Baseline data (V0), defined as the time of the switch to DTG + 3TC; the outcomes recorded were: anthropological and epidemiological variables, date of HIV diagnosis, viral load at diagnosis, CD4 and CD8 count at diagnosis, CD4 nadir, previous ART regimen, reason for ART change, history of adverse effects, previous virological failures, data from the HIV genotyping performed upon therapeutic failure if available and data from the routine blood and urine tests. Data were also collected from the scheduled appointments (V6, V24, V48) corresponding to follow-up visits performed at Week 6 ± 2 weeks, week 24 ± 8 weeks and week >48 after simplification.

Clinical and analytical data on any adverse effects during the follow-up visits were recorded. Adverse events were documented from the medical records. Special attention was paid to subjects who discontinued the study drug due to an adverse event, as well as cases with virological failures or detectable viral loads ≥50 copies/mL.

### 2.4. Pharmacoeconomic Study

A pharmacoeconomic analysis was performed comparing DTG + 3TC therapy versus BIC/TAF/FTC, DRV/c/TAF/FTC and RPV/DTG as the most widely used reference treatments in the current HIV panorama. To perform this analysis, we retrieved data from studies with patients with similar characteristics using the results obtained from this study in mITT patients for DTG + 3TC treatment and those from the systematic review performed by Mazzitelli et al. [17] for BIC/TAF/FTC, the EMERALD study for DRV/c/TAF/FTC [18] and the recent Rildo Study for RPV/DTG [19].

The costs of each treatment were calculated according to a list of laboratory sales prices consulted in February 2022, extracted from the coordinating hospital (VAT and state discounts not considered). These are shown in the summary table of the pharmacoeconomic study.

Cost minimization analysis was performed at one year to determine the saving generated by switching to DTG + 3TC. The cost-effectiveness ratio (CER) was also determined by dividing the cost of each treatment by its effectiveness, and the incremental CER (ICER) of DTG + 3TC therapy versus the others was calculated by dividing the difference in total costs (incremental cost) by the increase in effectiveness.

## 3. Results

### 3.1. Study Population and Patient Disposition

We included 200 patients with a median age of 51 years, 77.5% male with a median of 189 cells/µL of nadir CD4+ lymphocytes, diagnosed with HIV for a median of fifteen years. They had a median time on ART of 13 years, with a median of four previous ART regimens. Of these 96% had a baseline viral load <50 copies/mL and 708 cells/µL CD4+ lymphocytes on average. The main reasons for ART change were those related to toxicity or intolerance to the previous ART (42.5%) followed by ART simplification (39.5%) and to avoid drug-drug interactions (DDI) (13%) (for further details, see Table 1).

Of the initial 200 patients, 21 discontinued ART before week 24. Seven patients were withdrawn from treatment due to their physician’s decision (not related to pre-defined non-inclusion criteria), one due to their own decision to return to a standard triple therapy (3DR), and eight due to adverse effects.

Two unrelated deaths and three confirmed virological failures occurred.

Between weeks 24 and 48, 13 more patients discontinued ART: 3 were withdrawn from treatment due to their physician’s decision (not related to pre-defined non-inclusion criteria), one due to their own decision to return to a 3DR, one due to adverse effects and three lost to follow-up. Also, there were three unrelated deaths and two confirmed virologic failures.

There were three patients in whom after inquiring about previous HIV genotyping information, they had undergone previous HIV genotyping tests containing the 184V mutation. Although they were fully followed up, and maintained the viral suppression criterion, they were censored for the effectiveness analyses, as they met the preset non-inclusion criteria.

A total of 32 patients (16%) did not complete the 48-week follow-up from the baseline for reasons other than virologic failure.

Of the 168 patients who completed the follow-up at 48 weeks, a total of five VFs occurred, resulting in a 2.98% (5/168) VF rate (details in Figure 1).

### 3.2. Results per Subgroups

Upon analyzing these results in the previously explained subgroups we found the following:

(The 3 patients mentioned above with previous HIV genotyping with 184V were censored for these effectiveness analyses, therefore the N was reduced from 200 to 197 patients). Details in Figure 2.

ITT: In this subgroup of patients, which included all enrolled patients, at week 24 a total of three VFs were detected which is a percentage of 1.52% (3/197); moreover, 85.79% (169/197) were in virological suppression. At week 48, two more VF were detected, which amounts to a final value of 2.54% (5/197) of confirmed VF; meanwhile, 82.74% (163/197) of patients were in virological suppression.mITT: At week 24, we measured a VF rate of 1.67% (3/180), and 93.89% (169/180) of these were in virologic suppression. At week 48, we measured a VF rate of 2.78% (5/180), and 90.56% (163/180) of patients were in virologic suppression.PP: At week 24, we measured a VF rate of 1.74% (3/172), and98.26% (169/172) of these were in virologic suppression. At week 48, we measured a VF rate of 2.98% (5/168), and 97.07% (163/168) of patients were in virologic suppression.OD: This is the subgroup with virological data available for all patients during exposure to DTG + 3TC, which were collected from a total of 303 patients/year with 1.6% (5/303) having confirmed VF and 98.3% (298/303) of patients/year in virological suppression.

### 3.3. Therapeutic Outcomes

A total of 34 patients did not meet the study’s therapeutic success criteria at 48 weeks.

Five patients experienced virological failure, three of them were not tested and one underwent a genotypic study but did not amplify the sample. The latter was found to have the K103R and S147G mutations in the genotyping study (mutations that do not limit the activity of DTG [20,21]). All were subsequently monitored without significant problems changing their ART after confirmation of VF. None of them had recognized therapeutic adherence problems. (more details in Table 2).

There were seven blips in different patients and none had more than one blip. Four of them achieved resuppression without a change in ART, and three withdrew from the study due to their physician’s decision.

Five unrelated deaths occurred during follow-up: there were two deaths before 24 weeks and three deaths between 24 and 48 weeks. One of them with low viremia on last analysis (117 copies/mL). All causes of death were unrelated: one hepatic encephalopathy (decompensated cirrhosis), one sudden death due to severe atherosclerosis with stenosed vessels, one lung adenocarcinoma, one pulmonary epidermoid carcinoma and one esophageal carcinoma. (details are given in Table 3).

Nine instances of AEs were recorded: one of diarrhea, three alterations of the central nervous system (one of the patients withdrew before the week 4 analysis), two of asthenia, one of hypercholesterolemia, one drug interaction and one unspecified. Two patients decided to return to an STR, one before week 24 and one after (with VL = 51 copies/mL, no virological failure confirmed).

There were 10 discontinuations due to the decision of the physician, 7 before the 24-week follow-up visit and 3 after the follow-up visit. They were always unrelated to adverse effects or a lack of effectiveness, although two of them had low viremia at the time of drop-out (75 and 81 copies/mL without confirmation of virological failure).

There were three losses to follow-up, all occurring between weeks 24 and 48.

All patients who were withdrawn from the study due to AEs were reported to have mild-grade AEs and had a controlled viral load at the time of the switch (details in the Table 3).

Out of the 200 patients and after inquiring about previous HIV genotyping information, 91 (45.5%) had undergone a previous genotypic resistance test.

### 3.4. Biochemistry

The mean CD4+ lymphocytes increased significantly at 48 weeks (708.24 ± 323.4 cells/µL vs. 768.07 ± 322.55 cells/µL; *p* = 0.038). The CD4/CD8 ratio also increased from the baseline visit (0.86 ± 0.45 vs. 0.93 ± 0.45; *p* = 0.006). The mean creatinine levels increased by 0.09 mg/dL (1.03 vs. 1.12 mg/dL; *p* = 0.005) and the mean glomerular filtration rate decreased by 3.3 mL/min/1.73 m^2^ (86.55 vs. 83.25 mL/min/1.73 m^2^; *p* = 0.743). Regarding lipid profiles, the mean total cholesterol decreased (195 vs. 184 mg/dL; *p* = 0.004) with a significative change in the Castelli index (total cholesterol/HDL) (3.5 ± 1.65 vs. 4.05 ± 1.42; *p* = 0.009) and median triglycerides (127 [IQR 86–206] vs. 110.5 [IQR 76–157]; *p* = 0.0001). The remaining results can be found in Table 4 and Table 5.

### 3.5. Pharmacoeconomics

At the time of reviewing the results, the annual local costs per patient of DTG + 3TC, BIC/TAF/FTC, DRV/c/TAF/FTC and RPV/DTG are EUR 4851.24, EUR 6012.36, EUR 6094.56 and EUR 5538.96, respectively, so a switch to this dual therapy with DTG + 3TC would mean a potential annual saving per patient of EUR 1161.12 compared to BIC/TAF/FTC, EUR 1243.32 compared to DRV/c/TAF/FTC and EUR 687.72 compared to RPV/DTG in the Andalucía region (Table 6).

The virological efficacy ratios are 90.56%, 94.80% (13), 90.70% (14) and 97.50% (15), respectively, and, following the same order, the CERs are 54, 63, 67 and 57 indicating that the DTG + 3TC treatment is a cost-effective therapy for the treatment of HIV compared to the other therapies.

The ICER when comparing DTG + 3TC is 274 with BIC/TAF/FTC, −201 with DRV/c/TAF/FTC and 99 with RPV/DTG. These data suggests that DTG + 3TC therapy is cost-efficient compared to other therapies, as it provides slightly better results in terms of annual costs while maintaining appropriate virological suppression (Table 7). However, this does not mean that other therapies are not efficient alternatives for PLWHIV who may benefit from DTG + 3TC, as not all patients are optimal for treatment with this 2DR.

## 4. Discussion

The data obtained in this real-life study confirm that simplification to dual therapy with DTG + 3TC is an effective and safe as alternative to the standard triple therapy in pretreated and virologically stable HIV-infected patients, as shown in several clinical trials and other observational studies [14,15,16] along other dual therapies such as atazanavir/ritonavir plus lamivudine [5], dolutegravir plus boosted darunavir [22], rilpivirine plus boosted darunavir [23] or rilpivirne plus dolutegravir [19,24].

It should be noted that the patients in this cohort are different from those included in the simplification clinical trials, since they have more advanced HIV infection, associated with greater degrees of immunosuppression, and with greater exposure to ART (Table 1); these are characteristics that are usually non-inclusion criteria in simplification clinical trials, because they imply a greater risk of virological failure or increased difficulty in maintaining viral suppression.

A detailed reading of the baseline characteristics of these patients also helps us to visualize that in our setting, the combination of DTG + 3TC has been used among the population with polypharmacy (86%) and in situations of advanced disease (several previous ART regimens, older population, etc.) without relevant counterparts for the management of interactions. With 13% of participants in this cohort switching ART to avoid drug-drug interactions (DDIs), the results show that this 2DR could benefit this type of patient trying to avoid DDIs or managing co-morbidities, providing the inherent benefit of a safe and tolerable therapy.

We have to note that we detected eight patients with low level viremia (all of them <200 copies/mL) in the baseline viral load analysis at the moment of treatment change. Since these patients had an undetectable viral load in the previous six months, this event was not considered a reason for non-inclusion. All of these patients reached an undetectable viral load again in the next test and did not discontinue the treatment of the study, as per our standard of care procedure in those cases.

Even so, we found virological results similar to those achieved with triple therapy. The virologic failure rate in the per protocol analysis or the data observed analysis at week 48 is under 3% (being the virologic efficacy rate over 97%). Only five patients experienced virologic failure, and no treatment resistance mutations to DTG or 3TC were detected in the HIV genotyping study. However, due to the nature of this real-world study, we were unable to obtain a baseline genotype to properly assess the emergence of resistances. All of these patients were easily rescued and resuppressed.

Among the three patients (3.2%) with the 184V mutation, none experienced virological failure. The mutation was not known at the moment of data recording/patient inclusion and these patients were excluded from the primary efficacy analysis due to not meeting the inclusion/non-inclusion criteria.

Only seven instances of blips were observed, with no patients experiencing more than one. The highest viral load value recorded among these isolated instances was 239 copies/mL. In all cases, resuppression was achieved without discontinuing treatment, but by adding another antiretroviral in cases where treatment was discontinued.

The combination was very well tolerated, because there were only nine patients who withdrew due to adverse effects (4.5%), mild grade AEs in all cases (only three attributed to central nervous system alterations (33.34% of those)). If there were more discontinuations than expected (17%), they occurred due to reasons inherent to real-life retrospective multicenter observational studies, where the decisions of participating physicians and patients are not logically subject to strict, pre-established and homogenized criteria.

Regarding the long-term toxicity of ART and its impact on HIV-dependent residual inflammation, aging and the emergence of comorbidities, INSTIs (such as DTG) and 3TC have shown a favorable profile in comparison with other antiretrovirals [5,11,14,25,26,27]. Considering the need to monitor the long-term toxicity of ART throughout chronic treatment, it should also be noted that the use of DTG, which theoretically offers a better toxicity profile [28] and a possible favorable (or neutral) effect of INSTIs against the underlying chronic inflammation associated with HIV infection, [3], may offer a great advantage for these patients as demonstrated by various clinical trials [9,12] and our study.

Another noteworthy aspect of switching to this 2DR is that it significantly maintains an increase in CD4+ lymphocyte count (*p* = 0.038) and the CD4/CD8 ratio (*p* = 0.006) over time. In terms of changes in biochemical analysis, the improvement in the lipid profile achieved after switching to DTG + 3TC stands out as data consistent with the results of previously published clinical trials and studies with this combination of 3TC + DTG [9,11]. For details, refer to the significative changes after ART simplification table.

Our drug-economic study is based on the current scientific evidence available in pre-treated HIV patients with similar characteristics to the cohort studied. The results are similar to those of other studies performed with DTG + 3TC in these patients, although these results are not able to displace the other combinations as cost-effective options, especially since not all patients are candidates for this therapy. Our data support the DTG + 3TC combination, in appropriate patients, as the most cost-effective treatment, reducing the cost per patient/year.

These are the expected results of a combination that has effectively demonstrated its virological efficacy in multiple scenarios, and that has demonstrated a minimal toxic potential [24,25,26].

Our study has the limitations of an open-label, retrospective study. Regarding virological failure analysis, we do not have complete information on the evolution and consequences of the VFs. Out of the five VFs, only two genotypic studies were performed after failure and only one amplified. Although no follow-up problems were recorded or reported for these patients, there was a lack of supplementary data for the evolutionary analysis on rescue treatments and response. Therefore, no conclusions can be drawn in this respect. Regarding the emergence of new genotypic resistance, a follow-up period of 48 weeks may be considered short and could have left future drug resistance undetected. On the other hand, we found strengths in this study in that it is a multi-center study and offers data that support the results of clinical trials with real-world data.

## 5. Conclusions

The data from this study confirm the efficacy, safety and cost-effectiveness of simplification to 3TC + DTG in patients with well-controlled HIV infection and no history of virologic failure, which also improves their lipid profiles. It helps to visualize the idea that the potential benefit generated by reducing the number of antiretroviral drugs in the treatment of HIV infection may be greater than that expected from the prolonged maintenance of triple therapy.

## Figures and Tables

**Figure 1 viruses-16-00259-f001:**
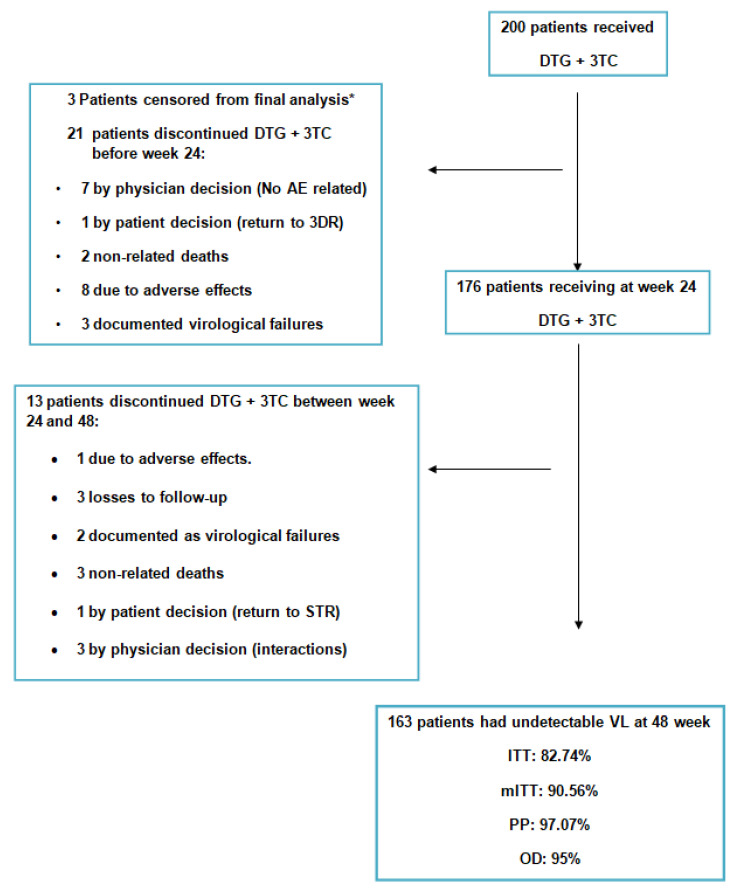
Global patient disposition over the study. * The 3 patients with the 184V mutation, not known at time of data recording. As they did not meet the inclusion/non-inclusion criteria, they were censored from the main objective analysis. DTG: dolutegravir; 3TC: lamivudine; AE: adverse effect; STR: single treatment regimen; VL: viral load; ITT: intention to treat; mITT: modified intention to treat; PP: per protocol.

**Figure 2 viruses-16-00259-f002:**
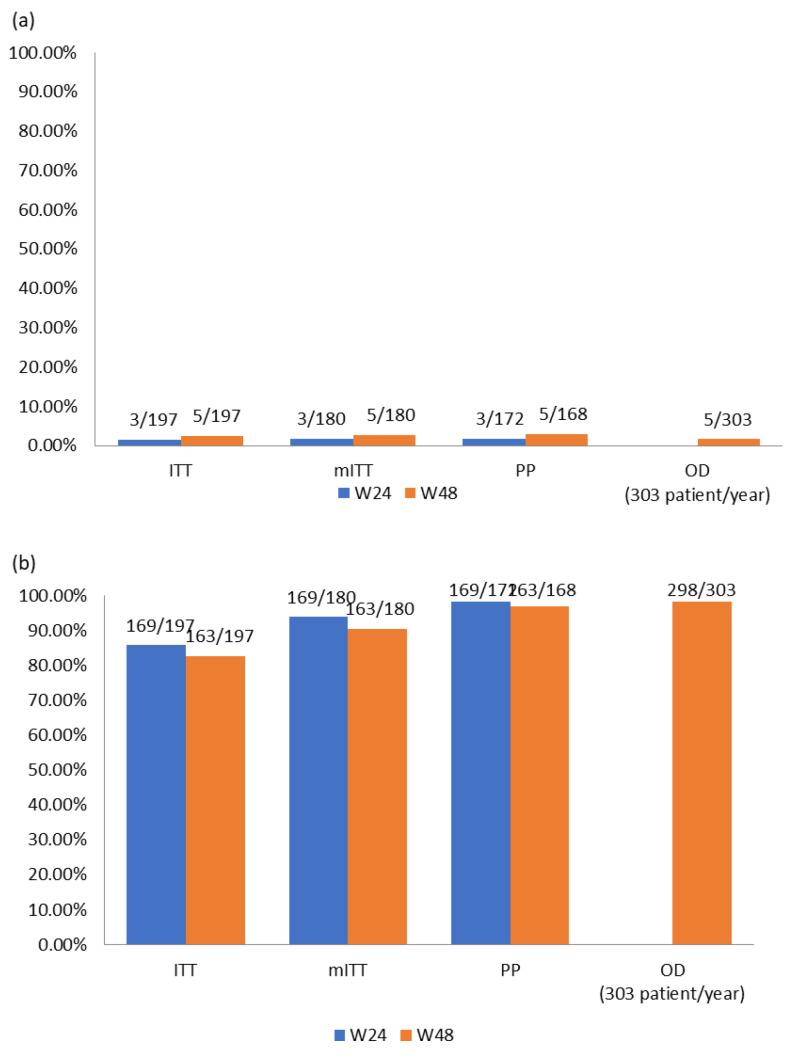
(**a**) Viral failure results (% of VF) and (**b**) viral suppression results (% of VL < 50 copies/mL).

**Table 1 viruses-16-00259-t001:** Baseline clinical characteristics.

N Total of Patients	*n* = 200
Age, years, median (IQR)	51 (44–56)
male, *n* (%)	155 (77.5%)
Time since HIV diagnosis, years, median (IQR)	15 (7–22)
CD4 nadir, median (IQR)	189 (59–309) cells/µL
Years of ART, median (IQR)	13 (4–18)
Previous ART regimens, median (IQR)	4 (2–8)
* Previous ART: *	
NRTI	156 (78%)
NNRTI	62 (32%)
PI	80 (40%)
INSTI	82 (41%)
Baseline ART was 3DR	106 (53.3%)
* Baseline viral load: *	
<50 copies/mL, *n* (%)	192 (96%)
≥50 copies/mL, *n* (%)	8 (4%)
CD4count, mean (±SD)	708.24 (±323.4) cells/µL
Polypharmacy (>1 basal treatment different from ART)	172 (86%)
* Reason for ART change *	
Toxicity/intolerance to the previous ART	85 (42.5%)
ART simplification	79 (39.5%)
Avoid drug-drug interaction	26 (13%)

IQR: Interquartile range; ART: antiretroviral therapy; NRTI: nucleoside reverse transcriptase inhibitor; NNRTI: non nucleoside reverse transcriptase inhibitor; PI: protease inhibitor; INSTI: integrase strand transfer inhibitors.

**Table 2 viruses-16-00259-t002:** VF details.

	Baseline ART _(Previous ART Regimens)_	Baseline VL (Copies/mL)	VL on VF (Copies/mL)	Week of VF	HIV Genotyping	Details (ART Change and More)
Patient 1	FTC + RAL + TDF (15)	64	200	52	Not performed	DTG + 3TC + ABV
Patient 2	RPV + DRV/b (10)	43	165	13	Not performed	No supplementary data, resupression confirmed
Patient 3	MRV + DRV/b (13)	36	576	6	Not performed	DTG + 3TC + RPV
Patient 4	ABC/3TC + RAL (5)	61	8899	18	K103R y S147G	3TC + DTG + DRV/b
Patient 5	3TC + DTG + RPV (2)	27	2330	25	Ddid not amplify	No supplementary data

None of the patients reported adherence problems.

**Table 3 viruses-16-00259-t003:** Adverse events and dropouts.

ADVERSE EVENTS AND DROPOUTS		Time of Occurence		
**AEs**		0–4 w	5–24 w	25–48 w
Central Nervous System	2 related to sleep problems 1 Depressive disorder	1	1	
				1
Metabolic	1 hipercholesterolemia		1	
Digestive	1 diarrhea	1		
Other	1 drug-drug interaction 1 unknown, but reported as AE 2 asthenia		1	
			1	
			1	1
**Non-related deaths**	1 hepatic encephalopathy (decompensated cirrhosis) 1 sudden death due to severe atherosclerosis with stenosed vessels 1 lung adenocarcinoma 1 squamous cell carcinoma 1 esophageal carcinoma (w. 117 copies/mL)			1
			1	
		1		
				1
		1		
**Virological failure**	5 already specified in Table 2		2	3
**Other dropouts**	10 physician decision 2 patient decision (return to STR) 3 lost to follow up	1	5	4
			1	1
				3

**Table 4 viruses-16-00259-t004:** Laboratory results.

	BASELINE (*n* = 200)	24 (*n* = 180)	48 (*n* = 166)	*p*
**CD4 Lymphocite Count _(cells/mm3)_(mean ± SD)**	708.24 ± 323.4	727 ± 323.75	768.07 ± 322.55	0.038
**CD4/CD8 (mean ± SD)**	0.86 ± 0.45	0.88 ± 0.4	0.93 ± 0.45	0.006
**Creatinine(mg/dL) (mean ± SD)**	1.03 ± 0.59	1.11 ± 0.67	1.12 ± 0.82	0.005
**Total Cholesterol (TC) (mg/dL) (mean ± SD)**	195.31 ± 51.31	191 ± 46.71	184.34 ± 44.59	0.004
**TC/HDL-C(mean ± SD)**	3.5 ± 1.65	3.92 ± 1.25	4.05 ± 1.42	0.009
**HDL-Cholesterol (mg/dL), median (IQR)**	53 (42–78)	49 (39–58)	47 (40–58)	0.002
**LDL-Cholesterol (mg/dL), median (IQR)**	98 (56–129)	110 (86–136)	109 (84–135)	0.005
**Triglicerides, median (IQR)**	127 (86–206)	121 (80–160.5)	110.5 (76–157)	0.0001
**GOT(IU/L), median (IQR)**	25 (20–32)	27 (21–35)	23 (19–29)	0.011
**GPT(IU/L), median (IQR)**	24 (17–36)	24 (18–39)	22.5 (16–34)	0.155
**GGT(IU/L), median (IQR)**	33 (20–59)	31 (17.2–48.7)	27 (19–44)	0.0001
**FA(IU/L), median (IQR)**	82.5 (59–98)	77 (62–93)	73.5 (58–91)	0.0001

**Table 5 viruses-16-00259-t005:** Laboratory results for different “*n*” (cont.).

	BASELINE	24w	48w	*p*
**Vitamin D (*n* = 20) (ng/mL), median (IQR)**	25.7 (17.8–31.5)	31 (19.8–36)	23.8 (19.7–29.8)	0.638
**Calcium (*n* = 48) (mg/dL), median (IQR)**	9.5 (9.2–9.9)		9.6 (9.1–9.8)	0.771
**Phosphorus (*n* = 46) (mg/dL), median (IQR)**	3.2 (2.7–3.5)		3.1 (2.8–3.5)	0.986
**Reactive C Protein (RCP) (*n* = 25) (mg/L)** **(mean ± SD)**	7.85 (±13.43)		2.26 (±2.25)	0.204

**Table 6 viruses-16-00259-t006:** Cost minimization analysis.

COST MINIMIZATION ANALYSIS OF DTG + 3TC	
**vs. BIC/TAF/FTC**	−EUR 1161.12
**vs. DRV/c/TAF/FTC**	−EUR 1243.32
**vs. RPV/DTG**	−EUR 687.72

DTG: dolutegravir; 3TC: lamivudine; vs.: versus; BIC: bictegravir; TAF: tenofovir alafenamide fumarate; FTC: emtricitabine; DRV/c: darunavir boosted with cobicistat; RPV: rilpivirine.

**Table 7 viruses-16-00259-t007:** Cost efficacy analysis.

COST-EFFICACY CER AND ICER				
ART	EFFICACY	EUR/YEAR	CER	ICER
**DTG + 3TC**	91%	EUR 4.851	54	-
**BIC/TAF/FTC**	95%	EUR 6.012	63	274
**DRV/c/TAF/FTC**	91%	EUR 6.095	72	−201
**RPV/DTG**	98%	EUR 5.539	57	99

DTG: dolutegravir; 3TC: lamivudine; vs.: versus; BIC: bictegravir; TAF: tenofovir alafenamide fumarate; FTC: emtricitabine; DRV/c: darunavir boosted with cobicistat; RPV: rilpivirine; ART: antiRetroviral treatment; CER: cost efficacy ratio; ICER: incresing cost efficacy ratio.

## Data Availability

The authors confirm the accuracy of the data provided for this study as well as its availability.

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
