# Peer review of "DOLAMA 200: Effectiveness and Safety of a Dual Therapy with Dolutegravir Plus Lamivudine in Treatment-Experienced HIV-1 Infected Real World Participants in Spain"

_viruses, 2024, doi:10.3390/v16020259_

Round 1

Reviewer 1 Report

Comments and Suggestions for Authors

Summary:

The manuscripts main goal is to use real world evidence to assess how the two drug DTG/3TC regimen compares with clinical study data generated in support of regulatory submissions, including in terms of antiviral efficacy as well as non-virologic effects potentially conferred by ARV toxicities and any residual & persistent HIV replication.

General concepting comments:

Article:

Overall the authors describe real world data that is useful to support other clinical work.  Ie, work the authors can provide here can add to and balance observations from Phase 3 studies for submission activities.

Title:  Effectiveness and safety of a dual therapy with Dolutegravir plus Lamivudine in treatment-experienced HIV patients

The title might be modified a bit to further detail, ie perhaps to include something about this study being for “...treatment experenced HIV-1 infected real world participants in Spain”

Abstract:

It would be good to include as “the aim of this study” that it was looking at patients with multiple previous ART regimen (median = 6) experience. (But reminds me, also see comments further on about apparent errors in the data reporting).

General considerations:

Needs overall extensive QC - eg typos and capitalization should be cleaned up, and there is quite a lot of unclear wording.  I have picked out some examples, but do not have time to fully QC/edit and provide needed suggestions for this manuscript.

Other Specific Comments:

Line 79:  virologic failure (RNA 79 HIV-1 > 50 copies/mL”.  Isn’t this actually “≥”?

Materials & Methods

Be consistent in use of "CD4/CD8" ratio (re “T4/T8" is also used).

Line 92:  Virological suppression maintained in the 6 months prior to the switch (1 blip allowed, defined as a  viral load ≥ 50 copies/mL followed by a retest result of Viral Load < 50 copies/mL) Who has been switched to DTG+3TC.   Grammatically poorly written.

Line 95:Patients with viral load ≥50 copies/mL detected in the baseline laboratory test results done at the moment of the switch will still be considered eligible if no change of the treatment was done.  This is unclear – the participants could have been on many different regimens (median = 6 according to the manuscript) over time, so there most likely will have been a change in treatment at some point.  Needs better definition.

Line 98: Exclusion criteria is poorly written; it has sentence fragments mixed with unclear sentence structure.

Line 108: Defineexitus letalis.

Line 119: Paragraph starting at 119 is poorly structured & needs to be rewritten.

Line 133: The first paragraph uses apparently the PROPOSED pharmacoeconomic analysis language, not what Was done.   Change the tense appropriately.

What resistance testing platform/methods/algorithm were used?

Results

Line 153:  “They had a median time on ART of 13 years, with a median of 4 previous ART lines.”   The introduction says median “..., twelve years on ART and 6 previous ART lines.”  Parts of either the introduction or results are incorrect.  Or is this a different population being analyzed?

Table 1 (line 158): is very hard to read, I can't tell which are headers easily - needs reformatting (not just center align and bullet some lines).

Many edits are needed in the Results section, eg, the mutations at line 209 have no identified reading frame, though it says the mutations do not limit integrase inhibitor activity.  And what algorithm is used to assess resistance?  That is needed.

Discussion

There are too many cases such as this sentence that is difficult to understand in this manuscript:  Only 7 blips were observed, no patient had more than 1 blip, the highest VL value reached in these isolated virological rebounds was 239 copies/mL, and in all cases re-suppression was achieved without withdrawing treatment but adding another antiretroviral in the cases that discontinued.

If this is noteworthy, it should be able to standalone to be understood.

Another example describing unclear data source, or &/or methods is at:

Line  301: “and there were only 5 patients with virologic failure, in whom no emergence of treatment resistance mutations was detected in the HIV genotyping study (despite one patient developing K103R y S147G mutations 303 as seen in table 2), and who were easily rescued and resuppressed.  What testing was done, Was it only for the integrase region?   Was resistance testing done with baseline samples? Comparison with baseline genotype is needed to assess emergent resistance.

References: seem appropriate and fairly comprehensive.

Comments on the Quality of English Language

Use of English and grammar needs fairly substantial review and revision.  The manuscript often uses jargon and too often is unclear – I did not have time to catalogue but some examples include eg, from Start of Abstract:

Line 16.  “and the increasingly understanding of...”  Use “Increasing” instead of “Increasingly”

Line 23.  “twelve years on ART and 6 previous ART lines.”   “lines” is jargon, use “regimens”.

Line 24.  “a total of 5 VF occurred, leaving a 2.98% (5/168) VF rate.”  Use something like “resulting in a 2.98% (5/168) VF rate” instead of “leaving

Introduction

Line 39.  “Although nowadays double therapy (2DR) is recognized as an elective option, even for naïve patients...”   Sentence as written in context of surrounding text should not start with “Although nowadays”

Line 40:  “2DR also could add several points in treatment 40 optimization”.  Use of “Points” is also jargon, sentence needs revision.

Line 60:  “...able to demonstrate, through multiple clinical trials, non-inferiority to triple therapy in practically all possible antiretroviral treatment scenarios.

The list of references does not cover “practically all possible antiretroviral scenarios” – eg not with pre-existing resistance, or known virological failure - briefly describe the actual studies types.

Line 77 – “and To determine de” – .  Such obvious typos should not exist.

Author Response

Thank you very much for your time and discussion on this manuscript. 

I have been able to incorporate changes to reflect most of the suggestions provided. I have highlighted the changes within the manuscript.

Here is a response to some comments and concerns:

  • Abstract:

It would be good to include as “the aim of this study” that it was looking at patients with multiple previous ART regimen (median = 6) experience. (But reminds me, also see comments further on about apparent errors in the data reporting).

  • Thank you for this suggestion. It would have been interesting to explore this aspect but looking at patients with multiple previous ART regimens was not the aim of our study at the time of design, and was what we found when analyzing the data.
  • Many edits are needed in the Results section, eg, the mutations at line 209 have no identified reading frame, though it says the mutations do not limit integrase inhibitor activity.  And what algorithm is used to assess resistance?  That is needed.
  • You have raised an important point. However, we have decided not to include that information in the manuscript as we believe it would not add relevant information. We have added clarifications and references to address any potential interest about it in the manuscript. Furthermore, we have requested additional details from the responsible person for this section in order to answer you: “HIV was sequenced using different protocols at testing sites, including the Sanger sequencing–based assay and next-generation sequencing (NGS)–based assay. For NGS sequences, a 20% consensus was generated and used for this study. The Stanford HIV database (version 9.1) was used for sequence alignment, quality assessment, resistance interpretation, and subtype assignment.”
  • Discussion

Another example describing unclear data source, or &/or methods is at:

Line  301: “and there were only 5 patients with virologic failure, in whom no emergence of treatment resistance mutations was detected in the HIV genotyping study (despite one patient developing K103R y S147G mutations 303 as seen in table 2), and who were easily rescued and resuppressed.”  What testing was done, Was it only for the integrase region?   Was resistance testing done with baseline samples? Comparison with baseline genotype is needed to assess emergent resistance.

  • Similarly to the previous comment, we have rephrased that section to provide further clarification.

In addition to the above comments, the manuscript was reviewed by
an experienced English-speaking colleague and all spelling and grammatical errors pointed out have been corrected.

Sincerely

Reviewer 2 Report

Comments and Suggestions for Authors

The article by Sequera-Arquelladas et al concerns the indication of dual therapy in people living with HIV who have been treated with different ART drugs for twelve years. The aim of the study is to evaluate the efficacy and safety of switching to doultegravir plus lamivudine. Efficacy will be assessed by virological failure with 48 weeks of follow-up for 200 people. The endpoint is therapeutic success at week 48. This study is retrospective and well written.
Minor comments
- There is a problem with the number of individuals, or perhaps the authors could explain this better. "...VF 23 occurred, leaving a 2.98% (5/168)" "....were a VF rate of 2.54% (5/197)" and "with viral suppression was 98.3% (298/303)".
- Instead of "Materials and methods", "Individuals and methods" would be more appropriate. Patients are not "materials".
- Exclusion could also be changed to non-inclusion.

Author Response

Thank you very much for your time and discussion on this manuscript. 

I have been able to incorporate changes to reflect most of the suggestions provided. I have highlighted the changes within the manuscript.

Here is a response to some comments and concerns:

  • Minor comments
    - There is a problem with the number of individuals, or perhaps the authors could explain this better. "...VF 23 occurred, leaving a 2.98% (5/168)" "....were a VF rate of 2.54% (5/197)" and "with viral suppression was 98.3% (298/303)".
  • The differences in the number of individuals in each analysis are described at the beginning of point 2.3 statistics and outcomes and correspond to different subsets of the same population. Answering the numbers you cited, 5/168 are the patients in the Per Protocol subset, 5/197 are the patients within the Intention To Treat subset and 298/303 are the Observed Data subset. Copiyng from the manuscript so you could read what population include or not-include each subset:

For the main objective analysis, patients were subset into 4 different subgroups: 1) intention-to-treat (ITT) analysis of all enrolled patients; 2) modified intention-to-treat (mITT) analysis, which excluded patients who withdrew from the study due to problems that altered the assessment of the predefined objectives such as physician decision, patient decision (both always unrelated to adverse effects) and not related deaths. AEs, VF and loss to follow-up maintained; 3) per protocol (PP) analysis, or patients who remained on the study combination until they reached 48 weeks or discontinued the study due to virologic failure or adverse effects. This subset will not include, in addition to mITT criteria, the loss to follow-up; and 4) observed data (OD) analysis, of available virologic data for all patients during exposure to DTG+3TC expressed in patient/year.

  • - Instead of "Materials and methods", "Individuals and methods" would be more appropriate. Patients are not "materials".
    - Exclusion could also be changed to non-inclusion.
  • We agree with this and have incorporated your suggestion throughout the manuscript.

Sincerely

Reviewer 3 Report

Comments and Suggestions for Authors

Please see the attached file with detailed comments. 

Comments on the Quality of English Language

please see the attached file

Author Response

Thank you very much for your time and discussion on this manuscript. 

I have been able to incorporate changes to reflect most of the suggestions provided. I have highlighted the changes within the manuscript.

Here is a response to some comments and concerns:

  • Abstract: please revise the intention to treat claim: those who were retained and have virologic suppression should be shown, which is 82.7%. It is not clear where the denominator of 303 (OD) could have come from considering that there were only 200 patients who received 3TC and DTG as dual therapy; please revise for clarity.
  • The abstract presents the study protocol's objectives, specifically the virological failure rate at 48 weeks, which is 2.54% in our ITT analysis. We preferred to include the OD analysis of virologic suppression over the ITT due to space restrictions of the journal. However, the ITT analysis of virologic suppression is shown in the results and in the graphs that it is indeed low. 
  • In our opinion, OD analysis is an appropriate method for assessing virological efficacy, and its application to any scenario (prospective studies, clinical trials, observational or retrospective studies...) does not cause it to lose homogeneity and power in the interpretation of results. The analysis of viremia evolution during exposure to experimental regimens (which we measure in terms of "patient years") in patients who have observable data, i.e. HIV viral load determinations, provides in our experience an appropriate way to assess the ability of any HAART to suppress viral replication, because it eliminates the biases that arise when introducing other variables that influence therapeutic success other than antiviral potency.
  • The 303 patients/year denominator in the observed data subset is achieved thanks to our collaborators who also provide information on the last viral load from these patients which maintains the study combination. This information provided data beyond our primary objective of 48 weeks. We believe that this data should be shared.
  • 2. Line 46: please differentiate residual viremia (detectable HIV-1 RNA below the threshold of ultrasensitive commercial viral load assays) from replication – residual viremia does not constitute replication as it is generally not suppressible (Dinoso et al. https://pubmed.ncbi.nlm.nih.gov/19470482/) – replication is defined as new rounds of replication – and would therefore imply evolution not the production of virions or the expression of viral protein (these replication steps are generally not suppressed by the actions of antiretroviral drugs)–– but several studies have shown that true replication and evolution is unlikely when viral loads are suppressed below the threshold of commercial assays (https://pubmed.ncbi.nlm.nih.gov/30310821/). Nevertheless, proteins can be expressed (https://www.ncbi.nlm.nih.gov/pmc/articles/PMC6374386/) and as is correctly pointed out could contribute to inflammation. In contrast, low level viremia -e.g., 50 – 200 copies/mL most often indicates ongoing replication. Perhaps the authors meant “low level viremia” instead of “residual viremia” – please clarify or correct.
  • Thank you for pointing this out. We agree with this comment (we meant residual viremia). Therefore, have rephrased that section to provide further clarification.
  • 3. Line 104: Please correct the statement – patients were “stratified”. Stratification is the division of participants into non-overlapping subgroups according to a participant characteristic. However this appears to refer to 4 different approaches to outcome analysis.
  • Agree. We have, accordingly, modified the terminology to "subset"
  • 4. Line 198: It is not understood how 303 patients could be included since only 200 were reported to be included in the study. Please clarify.
  • The response to this is provided in the first bullet point above.
  • 5. Line 294: Please revise the claim that the study had similar results to those achieved with triple therapy. Considering the biases introduced by the study design and confounders could impact on retention– nonrandomized the most important analysis is the intention to treat analysis of patients retained and in virological suppression which at 82.74% seems inferior to the Dawning and NADIA studies which included treatment experienced patients with treatment failure at baseline. This raises concern that the real-life effectiveness of dual therapy is relatively low.
  • Firstly, it should be noted that the use of Intention to Treat (ITT), Per Protocol (PP), and Observed Data (OD) analysis is primarily intended for interpreting results obtained in prospective studies. Translating these methods to observational and retrospective studies is always problematic, as you have pointed out. Additionally, the OD analysis is a reliable method for assessing virological efficacy, as previously stated.
  • Our cohort comprises a diverse range of patients, including those who have experienced treatment failure with previous ART, those with polypharmacy, and those who have undergone several previous ARV regimens.
  • When claiming similarity between our study and others, we consider all relevant studies, including those you mentioned. Upon examining the ITT analysis of the two clinical trials you mentioned, the DTG arms yielded results of 83.65% in the Dawning trial and 80.9% in the NADIA trial at VL< 50 copies/mL, which appear comparable to our own.
  • 6. Please add to the limitations the short follow-up as drug resistance may develop after 48 weeks when a regimen with low genotypic susceptibility score is used. Considering that many patients were treatment experienced and three more were discovered with M184V after enrolment, there may therefore be more patients with undetected or undisclosed drug resistance to either lamivudine or dolutegravir, compromising the longer-term durability of the regimen.
  • We agree with this and have incorporated your suggestion throughout the manuscript.

In addition to the above comments, the manuscript was reviewed by
an experienced English-speaking colleague and all spelling and grammatical errors pointed out have been corrected.

Sincerely